# Long-Term Clinical Outcomes of Left Atrial Appendage Closure in Patients with Left Atrial Appendage Thrombus

**DOI:** 10.3390/jcm14217589

**Published:** 2025-10-26

**Authors:** Moshe Katz, Rotem Nahmias Oz, Eias Massalha, Avi Sabag, Eyal Nof, Israel Barbash, Paul Fefer, Victor Guetta, Roy Beinart

**Affiliations:** 1Gray Faculty of Medical & Health Sciences, Tel Aviv University, Tel Aviv 6997801, Israelaias.masalha@sheba.health.gov.il (E.M.); avi.sabbag@sheba.health.gov.il (A.S.); eyal.nof@sheba.health.gov.il (E.N.); israel.barbash@sheba.health.gov.il (I.B.); paul.fefer@sheba.health.gov.il (P.F.); victor.guetta@sheba.health.gov.il (V.G.); roy.beinart@sheba.health.gov.il (R.B.); 2Cardiology Department, Sheba Medical Center, Tel-Hashomer, Ramat Gan 5262000, Israel

**Keywords:** atrial fibrillation, left atrial appendage, atrial thrombus, stroke, systemic embolism

## Abstract

**Background**: Patients with atrial fibrillation (AF) who have a high bleeding risk or contraindications to anticoagulation may be candidates for left atrial appendage closure (LAAC). However, the presence of a thrombus in the left atrial appendage (LAA) is generally considered a contraindication to the procedure. While the feasibility and short-term safety of LAAC in patients with pre-existing LAA thrombus has been reported, data on long-term outcomes remain limited. **Objective**: To assess the long-term clinical outcomes of AF patients undergoing LAAC in the presence of an LAA thrombus. **Methods**: This retrospective, single-center registry included all AF patients who underwent LAAC between June 2010 and April 2024. Patients were stratified based on the presence or absence of LAA thrombus at the time of the procedure. The primary endpoint was a 5-year composite of stroke, systemic embolism, or all-cause mortality. **Results**: A total of 403 patients underwent LAAC, of whom 24 (6%) had an LAA thrombus at the time of the procedure. During a median follow-up of 3.9 years, the primary endpoint occurred in 116 patients: 110 events (41%) in the no-thrombus group and 6 events (38%) in the thrombus group. There was no statistically significant difference in major adverse cardiovascular events (MACE) between groups (log-rank *p* = 0.862). **Conclusions**: LAAC may be performed safely in selected patients with distal LAA thrombus, with long-term outcomes comparable to those without thrombus.

## 1. Introduction

Atrial fibrillation (AF) is the most prevalent sustained cardiac arrhythmia, affecting over 46 million individuals globally as of 2016 [1]. Its incidence and prevalence rise sharply with age and given the global trend toward aging populations and increasing cardiovascular risk factors, the burden of AF is projected to grow substantially over the next few decades. In addition, AF is the leading cause of cardioembolic strokes, increasing the risk of ischemic stroke approximately five-fold [1]. About 20% of all ischemic strokes are cardioembolic in origin, and AF-related strokes are associated with higher morbidity and mortality compared to other ischemic strokes. Mortality can reach 37% within one year, with a recurrence rate of up to 17%. Therefore, stroke prevention remains a key therapeutic goal in AF management [2].

Two principal strategies exist for stroke prevention in AF: pharmacologic anticoagulation and mechanical closure of the left atrial appendage (LAA).

Warfarin has long been the standard therapy for preventing cerebrovascular events in atrial fibrillation, reducing ischemic stroke risk by about 60% [1]. However, its narrow therapeutic window, frequent food and drug interactions, and variability in metabolism make INR control difficult, limiting its use in many patients. These challenges prompted the development of direct oral anticoagulants (DOACs), which have proven non-inferior to warfarin in preventing ischemic stroke [3,4,5]. Nevertheless, the risk of major bleeding with those drugs appears to be similar to that of warfarin [6].

Accordingly, while oral anticoagulants are effective in reducing stroke risk, they are associated with significant bleeding risk and, at times, require careful monitoring. In patients with a history of major bleeding, such as spontaneous intracranial hemorrhage, or those with contraindications to long-term anticoagulation, transcatheter left atrial appendage closure (LAAC) may be considered as alternative to oral anticoagulation (OAC). Briefly, percutaneous LAAC involves the deployment of an occlusion device within the LAA. However, the presence of a pre-existing LAA thrombus is considered a contraindication due to the potential risk of thrombus dislodgement during catheter manipulation. The presence of left atrial thrombus in patients with non-rheumatic atrial fibrillation is a relatively common finding (17%) [7]. Therefore, emerging techniques and devices have been developed which allow LAAC to be performed in selected patients with LAA thrombus [8,9,10,11].

Current data regarding long-term outcomes in non-valvular atrial fibrillation (NVAF) patients undergoing LAAC in the presence of thrombus are limited. This study aims to evaluate the safety and long-term efficacy of LAAC in this high-risk subgroup.

## 2. Methods

### 2.1. Study Design

This study is based on a retrospective and prospective cohort, single-center registry of consecutive patients who underwent left atrial appendage closure procedures at the Sheba Medical Center from June 2010 to April 2024. The prospective cohort enrolled patients from February 2013 to April 2024. The retrospective cohort included patients that underwent LAAC procedure between June 2010 and February 2013.

Eligible patients included those with documented AF (CHA_2_DS_2_-VA score ≥ 1) and either a contraindication to long-term oral anticoagulation or a history of thromboembolic events despite adequate anticoagulation. Patients were excluded from the registry if they had proximal thrombus (≤15 mm from the left atrial appendage ostium) or had a thrombus floating in the left atrium (Appendix A).

Approval of data collection was obtained from the institutional review board, and all the patients in the prospective cohort provided informed consent for inclusion in the registry.

### 2.2. Thrombus Identification

Left atrial appendage thrombus was identified either during pre-procedural transesophageal echocardiography (TEE), computed tomography (CT) imaging, or on TEE performed at the outset of the LAAC procedure. In patients with LAA thrombus and no contraindications to long-term oral anticoagulation—such as those without a history of recurrent ischemic stroke—a therapeutic trial of low molecular weight heparin (LMWH) was initiated for a minimum of three weeks. Repeat TEE was then performed to assess for thrombus resolution. If the thrombus had been resolved, LAAC was performed in a standard manner to reduce thromboembolic risk. In cases where the thrombus persisted, LAAC was performed as a last-resort strategy for stroke prevention under carefully modified procedural precautions.

The study population was stratified into two subgroups: (1) patients without visible thrombus in the LAA at the time of the procedure (non-exposed group), and (2) patients in whom thrombus was still present in the LAA at the time of the procedure (exposed group).

### 2.3. Left Atrial Appendage Closure Procedure

All procedures were performed under general anesthesia using either first- or second-generation Amplatzer™ occluder devices (Cardiac Plug and Amulet; Abbott Vascular, Santa Clara, CA, USA) or Watchman™ and Watchman FLX™ devices (Boston Scientific, Marlborough, MA, USA). On the day of the procedure, all patients underwent echocardiographic evaluation—either TEE or intracardiac echocardiography (ICE)—to assess for the presence of LAA thrombus.

In cases where LAA thrombus was present at the implantation day, procedural techniques were modified from the conventional LAAC approach to minimize thromboembolic risk. These modifications included:Avoidance of contrast injection into the LAA prior to device deployment.Maintenance of the delivery sheath outside the LAA ostium before deployment.Cautious advancement of the device and sheath into the LAA only after partial deployment of the device—either the lobe in Amplatzer devices or the ball-shaped structure in Watchman devices to reach the intended landing zone (Figure 1A–C).

The above “no-touch” implantation technique was used to reduce the risk of thrombus dislodgement by avoiding intra-appendage contrast injection and minimizing catheter manipulation. In this technique, the device is preloaded in the left superior pulmonary vein and gently advanced to the LAA orifice, limiting contact with thrombotic material. This approach, often combined with cerebral embolic protection, has shown feasibility, safety, and high procedural success in patients with distal LAA thrombus [12,13,14].

The use of cerebral embolic protection (SENTINEL™; Boston Scientific, Marlborough, MA, USA) was left to the operator’s discretion (Figure 1D). Similarly, the choice and duration of post-procedural antithrombotic therapy were determined according to individual operator preference.

Before disconnecting the device, a “timed up and go” maneuver (TUG test) was performed under fluoroscopic guidance by gently pulling and pushing the delivery cable to ensure stable device positioning. Proper occlusion of the LAA orifice and the presence of any peri-device leaks were evaluated by echocardiography. Procedural success was defined as the complete absence of flow or the presence of a peri-device leak ≤ 5 mm, as assessed by color Doppler imaging.

### 2.4. Follow-Up and Endpoints

In-hospital follow-up was conducted through clinical examination and fluoroscopic evaluation of the implanted device on the day following the procedure to exclude device embolization. In cases of chest pain or hypotension, additional assessments, including electrocardiography and echocardiography—were performed to rule out complications.

Long-term follow-up was based on a comprehensive review of subsequent clinical records. Patients underwent TEE or CT imaging between 45 days and 3 months after the procedure to evaluate for residual peri-device leaks.

Procedural success was defined as the absence of any residual flow or the presence of only minimal peri-device flow (≤5 mm), as assessed by color Doppler on TEE or by LAA contrast-enhanced CT.

All bleeding events were classified according to the Bleeding Academic Research Consortium (BARC) criteria [9].

The primary endpoint of the study was a composite of all-cause mortality, stroke or transient ischemic attack (TIA), and peripheral systemic embolism at 5-year follow-up.

## 3. Statistical Analysis

Comparisons between groups were analyzed by Chi-square or Fisher’s exact tests for categorical parameters, and Student’s *t* test or Mann–Whitney test for continuous parameters, as appropriate. Non-normal distributed continuous variables were reported as median and interquartile range [Q1–Q3] and normally distributed continues variables were reported as mean ± 1 SD. Categorical variables were described in terms of counts and percentage. The length of follow up was described using reverse censoring method. Kaplan–Meier curves were used to describe time to the primary end point. A Log Rank test was used to compare survival between groups. Cox regression was performed to estimate the ratio for the primary endpoint. A multivariate Cox regression analysis was performed to evaluate possible confounders.

All statistical tests were two-tailed. *p* < 0.05 was considered statistically significant. All statistical analyses were performed using SPSS 29.0 (IBM, Armonk, NY, USA).

## 4. Results

Between June 2010 and April 2024, 408 consecutive patients underwent LAAC at the Sheba Medical Center. Five patients were excluded from the study. One out of the five was excluded due to an appendage that was too small to accommodate available devices (Figure 2). Final cohort included 403 patients. Of these, 24 patients (6%) had an LAA thrombus visualized at the time of the procedure (Figure 2). Comparison of baseline characteristics between patients with LAA thrombus at implantation day and patients without a LAA thrombus is shown in Table 1. The indications for LAAC are shown in Figure 3. The most common indication was uncontrolled bleeding source (49%) followed by history of intracranial bleeding (37%).

The median patient age was 76 years (IQR [71–82]), with 65% being male. Paroxysmal AF was the predominant arrhythmia type (59%).

Among 44 patients, LAA thrombus was detected by TEE or CT prior to the procedure. Of them, 36 patients were given intensified anticoagulation, and thrombus resolution was observed in 26 (72%), enabling standard LAAC (Figure 2). The remaining 10 patients (28%) had persistent thrombus despite intensification of anticoagulation and underwent LAAC using ‘no-touch’ approach, comprising 42% of patients undergoing ‘no touch’ technique. Eight patients in whom a thrombus was identified before the procedure underwent direct LAAC because they had contraindication to anticoagulation. Interestingly, among patients with a left appendage thrombus, a larger LA diameter was found 4.5 ± 0.7 vs. 4.8 ± 0.6, *p* = 0.027, though LAA morphology were like patients without thrombus (Table 2). Amulet was utilized in 9 patients (38%) with LAA thrombus and Watchman Flex in the other 15 patients (62%) (Table 2).

Amulet and Watchman FLX devices were utilized in thrombus patients, while older-generation devices (Watchman, Amplatzer Cardiac Plug) were used in non-thrombus cases. Cerebral embolic protection (Sentinel) was used in 21% of thrombus patients.

Procedural success was evaluated by TEE or CT imaging between 45 days and 3 months post-implantation to assess residual peri-device leaks. Procedural success was achieved in 401 out of 403 patients (99.5%). Procedural success was achieved in 24 out of 24 patients with LAA thrombus (100%). Among patients with residual peri-device leaks, two had leaks > 5 mm, 13 had leaks of 3–5 mm, and 24 had leaks of 1–3 mm.

No periprocedural stroke occurred in the thrombus group (Table 3). One periprocedural stroke occurred in the non-thrombus group and was attributed to critical carotid stenosis. Three deaths occurred in non-thrombus patients, two of which were caused by device embolization: in one patient the device dislodged towards the mitral valve. Papillary muscle rupture occurred during percutaneous retrieval of the device and resulted in severe mitral regurgitation. The patient underwent an emergency operation but died on the same day; In the other patient, device embolization to the left ventricle led to cardiac arrest, and resuscitation failed. The third patient had severe chronic obstructive pulmonary disease (COPD), was readmitted with pneumonia 30 days following the procedure, and died from respiratory failure. Periprocedural death occurred in one patient with LAA thrombus. This patient underwent emergent LAAC and mitral clip implantation due to cardiogenic shock but eventually died from septicemia.

Most patients without LA thrombus were discharged on DAPT, while only third of the patients with a thrombus were discharged on DAPT (*p* = 0.011). However, more patients in the thrombus group received anticoagulation at discharge (33% vs. 10%, *p* = 0.003).

Major bleeding (BARC 3–5) occurred in one patient with LA thrombus (4.2%) and in 7 (1.8%) non-thrombus patients (Table 3). Among these patients, three experienced bleeding while on Clopidogrel, four while on aspirin and Clopidogrel, and one while on Warfarin and aspirin. Two major bleeding events were as direct result of procedure complication (vascular and upper airway injury). Six major bleeding events were gastrointestinal bleedings (GI), 4 out of the six were on dual antiplatelet therapy (DAPT) and one patient was one warfarin and aspirin. One patient suffered from GI bleeding while on Clopidogrel, but he had chronic thrombocytopenia as well (Table 4).

Vascular complications were significantly more common in the thrombus group (12.5% vs. 2.1%, *p* = 0.022). During follow-up similar bleeding events occurred in both groups (8.2% vs. 8.3%, *p* value = 1) (Table 5) although the antiplatelet and anticoagulant regimens at discharge were significantly different.

During a median follow-up of 3.9 years (IQR [3.7–4.1]), the composite primary endpoint (stroke, systemic embolism, or all-cause mortality) occurred in 116 patients. Event rates were similar between groups: 41% in the non-thrombus group vs. 38% in the thrombus group (log-rank *p* = 0.862) (Figure 4).

Multivariate Cox regression analysis identified age, congestive heart failure, nursing home dependency, and impaired renal function as significant predictors of MACE (Table 6). The presence of LAA thrombus was not associated with worse outcomes. Interestingly, no systemic embolic events were observed in either group during follow-up. No strokes occurred in the exposed group, whereas 19 stroke events were reported among patients without visible thrombus. Among those who experienced stroke, approximately half were discharged on dual antiplatelet therapy (DAPT), 40% on single antiplatelet therapy (SAPT), and 10% received no antiplatelet therapy. Regarding anticoagulation, 15% of these patients were treated with a direct oral anticoagulant (DOAC), 10% with low-molecular-weight heparin (LMWH), and 5% with warfarin.

## 5. Discussion

This study provides long-term clinical outcome data for patients with AF undergoing left atrial appendage closure (LAAC) in the presence of a thrombus within the left atrial appendage (LAA), a clinical scenario traditionally regarded as high-risk and often excluded from percutaneous intervention. While previous studies have mostly focused on the feasibility and short-term safety of LAAC in this population, our registry adds new evidence by providing long-term follow-up of up to five years.

LAAC has emerged as an effective strategy for stroke prevention in patients with non-valvular atrial fibrillation (NVAF), particularly in those with contraindications to long-term oral anticoagulation. However, the presence of an LAA thrombus is generally considered a contraindication due to the procedural risk of embolization. The presence of atrial thrombi has been excluded in randomized LAAC clinical trials [8,15,16,17]. To mitigate this, various procedural adaptations have been proposed, including the thrombus trapping technique (TTP-LAAC) [10] and the ‘no-touch’ technique [14], both of which avoid manipulation of catheters or angiography in the LAA.

In our cohort, device implantation was successful in 99.5% of cases, comparable to success rates of large multicenter studies [18,19]. In patients with LAA thrombus we achieved 100% success rate.

The TRAPEUR registry [13], a multicenter study of over 1900 patients, demonstrated the feasibility and short-term safety of LAAC in patients with LAA thrombus. Among 53 patients treated with modified techniques, only one ischemic event occurred within 30 days. Our results are consistent with these findings, as none of the patients with thrombus in our study experienced periprocedural stroke or embolism. In addition, we did not encounter any pericardial effusion/tamponade or device embolization in this high-risk cohort. We confirm that performing the “no touch” technique in patients with distal LAA thrombus by an experienced operators is safe and feasible. The higher rate of vascular complications in the thrombus group (12.5%) in our cohort may be attributed to more intensive periprocedural anticoagulation and inconsistent use of ultrasound guidance for vascular access. Performing real-time ultrasound guidance of femoral veins routinely may reduce vascular complications and inadvertent artery puncture [20]. Therefore, in recent years, we have started using ultrasound routinely to reduce vascular complications.

Another interesting finding in our study was the high successful rate of LAA thrombus resolution by anticoagulation intensification. Our findings support the role of anticoagulation intensification in cases where LAA thrombus is identified prior to the planned LAAC. In our cohort, 36 patients were treated with intensified anticoagulation (typically low molecular weight heparin), resulting in thrombus resolution in 72% of cases. This approach aligns with prior studies [10,14] and supports the idea that early thrombus resolution may permit safer and more conventional LAAC procedures. Marroquin et al. showed in their observational study that intensification of antithrombotic therapy (IAT) was the preferred strategy for LAA thrombus management over direct LAAC. Thrombus resolution was observed in 60% and 75% after initial and subsequent IAT, respectively. Similarly to our study, parenteral anticoagulation was the most common IAT strategy. They did not find statistically significant difference between IAT group and LAAC group in the primary endpoint of composite of bleeding, stroke and death at 18 months [11]. IAT was usually chosen where the thrombus was located proximal to the appendage and in cases where mobile thrombus was identified.

Nonetheless, some patients have absolute contraindication for anticoagulation and therefore cannot receive anticoagulation intensification, and some patients fail the IAT strategy and have recurrent stroke despite anticoagulation. In these cases, we employed contemporary devices (Amulet and Watchman FLX) known for their controlled and ostial deployment, reducing the need for deep catheter manipulation. These devices offer safer profiles in thrombus-bearing appendages, as has been previously suggested [7,8]. In our center, occluder device selection for LAAC procedures is performed randomly rather than according to patient or anatomical characteristics. This approach helps to minimize selection bias and potential confounding arising from device type. Device implantation was not feasible only in one patient due to unfavorable anatomy.

Additionally, the use of embolic protection device (EPD) (Sentinel™ Cerebral Protection System, Boston Scientific. Marlborough, MA) in selected patients likely contributed to the favorable safety profile observed. Boccuzzi et al. [11]. demonstrated that such protection can capture embolic debris released during LAA procedures. In our cohort, 21% of thrombus patients received cerebral protection, particularly when the thrombus was mobile or located near the appendage ostium.

Importantly, our study provides five-year outcome data, filling a critical gap in the literature. Most prior reports provided only 30-day or short-term outcomes [7,10,11]. The Kaplan–Meier analysis in our registry demonstrates that patients with thrombus had a similar rate of major adverse cardiovascular events (MACE) compared to those without thrombus (38% vs. 41%, respectively; *p* = 0.862). These results reinforce that, when performed under controlled conditions, LAAC in the presence of LAA thrombus is feasible and has excellent long-term results.

At discharge, the antithrombotic and antiplatelet regimens were statistically significant difference between groups. Anticoagulation was prescribed more for patients who underwent LAAC in the presence of a thrombus, while DAPT was prescribed more for patients without a thrombus. Nevertheless, no systemic embolic events were observed in either group during follow-up. During follow-up, strokes occurred exclusively among patients in the non-exposed group. The antithrombotic regimen prescribed in the non-exposed group was comparable to that used in the EWOLUTION study [21], thereby supporting the external validity of our findings. However, variations in the post-procedural antithrombotic therapy between groups may have influenced thromboembolic outcomes. The low incidence of systemic embolism and stroke limited our ability to perform further stratification or sensitivity analyses.

Only 2 bleedings events were as a direct result of the procedure. Most of the bleeding events in the first 30 days after the procedure were because of recurrence of gastrointestinal bleeding while on DAPT. A systematic review and meta-analysis comparing single antiplatelet therapy (SAPT) versus DAPT following LAAC showed that minimalistic strategy with SAPT did not significantly differ from DAPT regimens regarding the rate of stroke, device-related thrombosis and major bleeding [22]. Indeed, during follow-up, bleedings events were similar in both groups (8.2% vs. 8.3%, *p* value = 1).

Our findings must be interpreted within the context of a tertiary center where procedural expertise and device selection play a critical role in outcomes. Nonetheless, they provide meaningful real-world evidence suggesting that LAA thrombus should not be considered an absolute contraindication to LAAC.

In conclusion, LAAC may be considered in selected patients with LAA thrombus when thrombus is localized, anticoagulation is contraindicated or ineffective, and the procedure is performed with appropriate precautions. Techniques such as the thrombus trapping method, the use of advanced devices, and selective use of cerebral protection appear to enable safe and effective outcomes. Future multicenter studies and randomized trials are needed to further define patient selection criteria, validate procedural strategies, and establish evidence-based protocols for safely managing LAA thrombus during LAAC.

## 6. Limitations

This study is limited by its retrospective and non-randomized design. Selection bias is possible, as some patients with thrombus may not have been offered LAAC due to perceived high procedural risk.

Our thrombus cohort was relatively small (n = 24), which limits statistical power. The absence of significant differences in MACE should be interpreted with caution, as it may be attributed to insufficient statistical power rather than an actual equivalence between groups. Additionally, follow-up imaging was not available in all patients, and some procedural variables may have been under-reported.

Despite these limitations, this registry provides important long-term outcome data from a high-volume center and supports the consideration of LAAC in selected thrombus patients.

## Figures and Tables

**Figure 1 jcm-14-07589-f001:**
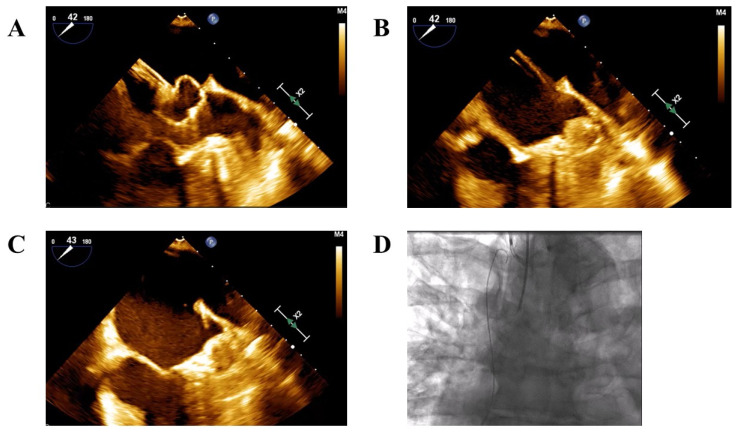
‘No-touch’ technique under transesophageal guidance. (**A**). The Watchman Flex device is positioned outside the left atrial appendage ostium, where it expands into a ball-like structure. (**B**). The Watchman Flex is advanced cautiously into the appendage to trap the left atrial appendage thrombus. (**C**). The device is deployed in the appendage with optimal anchoring and sealing. (**D**). Cerebral embolic protection with the Sentinel device (Boston Scientific) was established before performing left atrial appendage closure.

**Figure 2 jcm-14-07589-f002:**
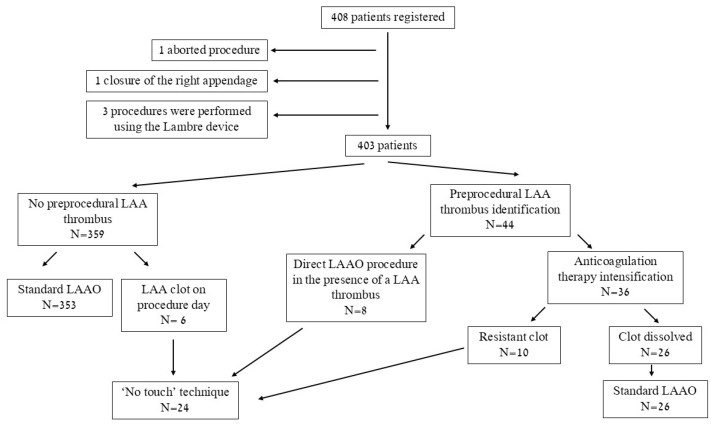
Study Population Flowchart and procedural Stratification Based on Presence of Left Atrial Appendage Thrombus.

**Figure 3 jcm-14-07589-f003:**
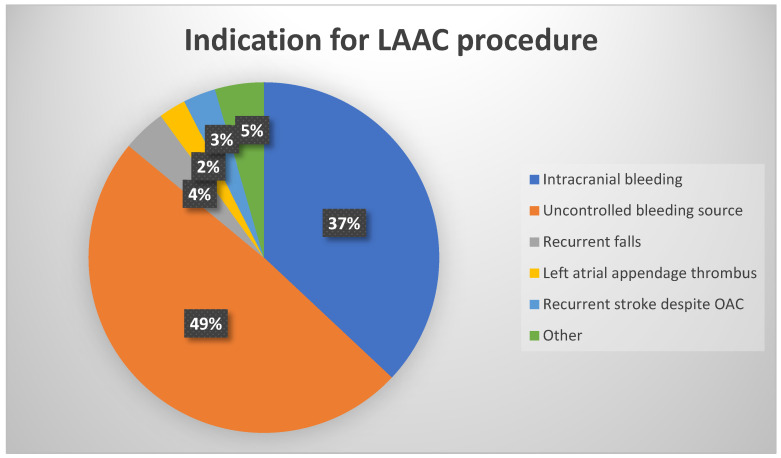
Indications for Left Atrial Appendage Closure in the study population.

**Figure 4 jcm-14-07589-f004:**
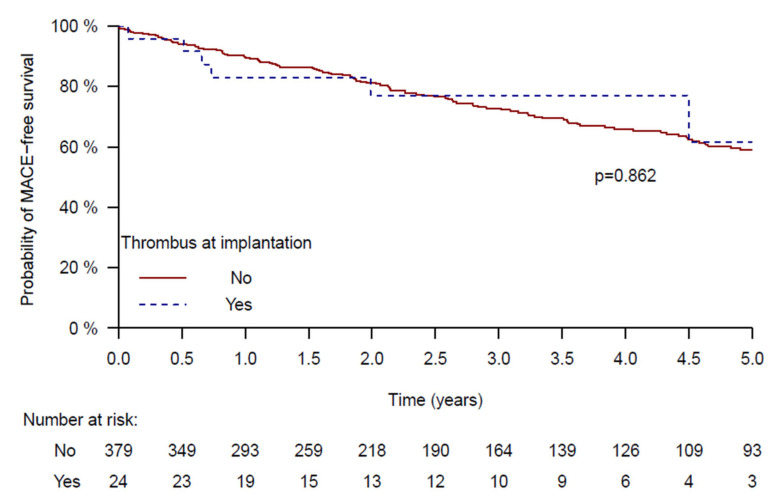
Kaplan–Meier Curve for 5-Year MACE in patients Undergoing LAAC With vs. Without Left Atrial Appendage Thrombus.

**Table 1 jcm-14-07589-t001:** Baseline demographic characteristics and co-morbidities.

	Overall N = 403	No LA Thrombus N = 379	LA Thrombus N = 24	*p*-Value
Age, Years	76 [71–82]	76 [71–82]	80 [75–85]	0.054
Sex, Male	260 (65)	242	18	0.268
Hypertension, mmHg	343 (85)	324 (86)	19 (79)	0.379
Active smoker	39 (10)	38 (10)	1 (4)	0.605
AF pattern *				0.109
Paroxysmal	237 (59)	227 (60)	10 (44)	
Persistent	62 (15)	59 (16)	3 (13)	
Permanent	100 (25)	90 (24)	10 (43)	
CHA2DS2VA	4 [3–5]	4 [3–5]	5 [3–6]	0.446
CHF	110 (27)	102 (27)	8 (33)	0.494
Diabetes mellitus	194 (48)	183 (48)	11 (46)	0.816
Dyslipidemia	273 (68)	258 (68)	15 (63)	0.571
CKD	129 (32)	124 (33)	5 (21)	0.226
Carotid artery disease	33 (8)	32 (8)	1 (4)	0.709
PVD	69 (17)	65 (17)	4 (17)	1
Ischemic heart disease	176 (44)	166 (44)	10 (42)	0.838
Permanent pacemaker	74 (18)	70 (19)	4 (17)	1
Cognitive impairment	44 (11)	40 (11)	4 (17)	0.32
Malignancy	38 (9)	35 (9)	3 (13)	0.756
Previous stroke	139 (35)	128 (34)	11 (46)	0.228
Previous hemorrhagic stroke	88 (22)	85 (22)	3 (13)	0.254
Previous intracranial bleeding	129 (32)	124 (33)	5 (21)	0.226
Previous GI bleeding	162 (40)	154 (41)	8 (33)	0.479
Other bleeding source	78 (19)	71 (19)	7 (29)	0.283
Recurrent falls	37 (9)	36 (10)	1 (4)	0.713
BMI, kg/m^2^	27 [25–30]	27 [25–30]	26 [24–29]	0.426
Hemoglobin (g\dL)	12 ± 1.9	11.9 ± 1.9	12.2 ± 1.6	0.457
Platelets (×10^3^/µL)	184 [147–233]	183 [146–231]	205 [170–264]	0.115
Creatinine (mg\dL)	1.11 [0.88–1.41]	1.12 [0.89–1.41]	0.97 [0.77–1.35]	0.179
eGFR (mL/min/1.73 m^2^)	63 [45–80]	61 [45–80]	72 [48–90]	0.144

Data are mean ± SD for normally distributed continuous variables and median [interquartile range] for non-normally distributed variables. Categorial variables are presented as number (percentage). Missing data is presented with *. AF—atrial fibrillation; LA—left atrium; CHF—congestive heart failure; CKD—chronic kidney disease; PVD—peripheral vascular disease; GI—gastrointestinal; BMI—body mass index; eGFR—estimated glomerular filtration rate.

**Table 2 jcm-14-07589-t002:** Procedural characteristics.

	Overall N = 403	No LA Thrombus N = 379	LA Thrombus N = 24	*p* Value
LAA device size, mm	27 [24–28]	27 [24–28]	27 [25–30]	0.248
LVEF, %	60 [50–60]	60 [50–60]	55 [35–60]	0.080
LVEDD, cm	4.7 [4.4–5.2]	4.7 [4.4–5.2]	4.85 [4.4–5.5]	0.374
LVESD, cm	3.1 [2.7–3.6]	3.1 [2.7–3.6]	3.1 [2.8–4.1]	0.503
LA diameter, cm	4.5 ± 0.7	4.5 ± 0.7	4.8 ± 0.6	0.027
LA area, cm	26 ± 6.4	26 ± 6.4	28.9 ± 6.8	0.054
LAA morphology *				0.267
Chicken wing	129 (33)	122 (33)	7 (32)	
Cactus	27 (7)	27 (8)	0	
Cauliflower	57 (15)	56 (15)	1 (5)	
Windsock	61 (16)	55 (15)	6 (27)	
Other	115 (29)	107 (29)	8 (36)	
Device type				0.003
AGA	23 (6)	23 (6)	0	
Amulet	204 (50)	195 (51)	9 (38)	
Watchman	56 (14)	56 (15)	0	
Watchman Flex	120 (30)	105 (28)	15 (62)	

Data are mean ± SD for normally distributed continuous variables and median [interquartile range] for non-normally distributed variables. Categorial variables are presented as number (percentage). Missing data is presented with *.

**Table 3 jcm-14-07589-t003:** Periprocedural complications.

	No LA Thrombus N = 379	LA Thrombus N = 24	*p*-Value
Death	3 (0.8)	1 (4.2)	0.219
Stroke	1 (0.3)	0	1
Peripheral embolism	0	0	NA
Pericardial effusion	2 (0.5)	0	1
Tamponade	3 (0.8)	0	1
Minor bleeding (BARC 1–2)	9 (2.4)	1 (4.2)	0.463
Major bleeding (BARC 3–5)	7 (1.8)	1 (4.2)	0.391
Vascular complications	8 (2.1)	3 (12.5)	0.022
Device embolization	4 (1.1)	0	1

NA—not applicable.

**Table 4 jcm-14-07589-t004:** Periprocedural major bleeding events.

Group	Procedure Indication	Antiplatelets/Anticoagulants Regimen at Discharge	Bleeding Event
No LA thrombus	GI bleeding	DAPT	Gastrointestinal
No LA thrombus	Intracranial bleeding	SAPT	Vascular injury
No LA thrombus	GI bleeding	DAPT	Gastrointestinal
No LA thrombus	GI bleeding	SAPT	Upper airway
No LA thrombus	GI bleeding and thrombocytopenia	SAPT	Gastrointestinal
No LA thrombus	GI bleeding	DAPT	Gastrointestinal
No LA thrombus	GI bleeding	DAPT	Gastrointestinal
LA thrombus	Recurrent stroke	Warfarin and SAPT	Gastrointestinal

LA—left atrium; DAPT—dual antiplatelet therapy; SAPT—single antiplatelet therapy.

**Table 5 jcm-14-07589-t005:** Antiplatelet and anticoagulation medications at discharge and bleeding events during follow-up.

	No LA Thrombus N = 379	LA Thrombus N = 24	*p*-Value
Antiplatelet therapy			0.011
Single antiplatelet	143 (38)	12 (50)	
Dual antiplatelet	218 (58)	8 (33)	
No antiplatelet	18 (4)	8 (17)	
Anticoagulation			0.003
Warfarin	10 (3)	1 (4)	
LMWH	10 (3)	3 (12)	
DOAC	18 (4)	4 (17)	
Bleeding events	31 (8.2)	2 (8.3)	1

LMWH—low molecular weight heparin; DOAC—direct oral anticoagulant.

**Table 6 jcm-14-07589-t006:** Multivariate Cox regression analysis.

Covariate	HR	95% CI for HR	*p*-Value
LAA thrombus	0.79	0.27–2.27	0.658
Sex, Female	0.75	0.68–1.73	0.747
Age, Years	1.03	1.00–1.07	0.039
Congestive heart failure	1.63	1.03–2.6	0.039
Nursing assistant	1.87	1.15–3.07	0.012
eGFR	0.988	0.978–0.999	0.029

LAA—left atrial appendage; eGFRx–estimated glomerular filtration rate.

## Data Availability

Data supporting the findings of this study are available from the corresponding author upon reasonable request.

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
