# Peer review of "Long-Term Clinical Outcomes of Left Atrial Appendage Closure in Patients with Left Atrial Appendage Thrombus"

_jcm, 2025, doi:10.3390/jcm14217589_

Round 1
Reviewer 1 Report
Comments and Suggestions for Authors
The study addresses an underexplored area—LAAC in patients with pre-existing thrombus. While the long-term outcomes add value, the novelty is somewhat diminished given prior short-term reports. The following points should be addressed:
-The thrombus group is small (n=24, 6% of the cohort). This severely limits statistical power, especially for rare outcomes such as stroke or systemic embolism. The authors should acknowledge more explicitly that the absence of statistical differences may reflect underpowering rather than true equivalence.
-As a retrospective, single-center study, selection bias is inevitable. It remains unclear how patients with visible thrombus but deemed too high-risk for LAAC were managed. A flow diagram of screened versus included patients would clarify the representativeness of the cohort.
-Device types (Amulet vs Watchman vs Watchman FLX vs older ACP) and procedural strategies (no-touch technique, cerebral embolic protection) varied substantially between groups. These differences are potential confounders and should be adjusted for or at least discussed in more detail. Outcomes may relate as much to operator/device selection as to thrombus presence.
-Post-procedural antithrombotic therapy differed significantly between groups. Given the centrality of bleeding and thromboembolism outcomes, this heterogeneity should be addressed with stratified or sensitivity analyses.
-The composite endpoint includes all-cause mortality, which is highly influenced by comorbidities unrelated to thromboembolic protection. The authors should provide cause-specific mortality breakdowns and consider presenting stroke/systemic embolism separately.
Author Response
Comments 1:
The thrombus group is small (n=24, 6% of the cohort). This severely limits statistical power, especially for rare outcomes such as stroke or systemic embolism. The authors should acknowledge more explicitly that the absence of statistical differences may reflect underpowering rather than true equivalence.
Response 1
We thank the reviewer for this valuable comment.
We acknowledge that the non-significant difference in MACE could be due to the study being underpowered to detect differences, rather than indicating genuine clinical equivalence between the groups.
Accordingly, we have revised the limitation section (page 14, lines 366-369) to clarify that the absence of statistical significance may reflect limited statistical power rather than true clinical equivalence between the groups. The revised text now reads:
The absence of significant differences in MACE should be interpreted with caution, as it may be attributed to insufficient statistical power rather than an actual equivalence between groups.
Comments 2:
As a retrospective, single-center study, selection bias is inevitable. It remains unclear how patients with visible thrombus but deemed too high-risk for LAAC were managed. A flow diagram of screened versus included patients would clarify the representativeness of the cohort.
Response 2
We thank the reviewer for this insightful comment. We fully agree that selection bias is inherent to our study design. To improve transparency, we have now added a detailed description of patients excluded from our registry in the Methods section (page 3, lines 94-96).
In addition, a flow diagram (Supplemental) has been included to illustrate the screened patients, excluded, and included in the final analysis.
Specifically, patients with visible LAA thrombus who were deemed too high-risk for LAAC were managed conservatively with optimized medical therapy according to their ability to receive anticoagulation as shown in the flow diagram.
Comments 3:
Device types (Amulet vs Watchman vs Watchman FLX vs older ACP) and procedural strategies (no-touch technique, cerebral embolic protection) varied substantially between groups. These differences are potential confounders and should be adjusted for or at least discussed in more detail. Outcomes may relate as much to operator/device selection as to thrombus presence.
Response 3
We thank the reviewer for this important observation. We agree that variations in device type and procedural strategy (no-touch technique, use of cerebral embolic protection) represent potential confounding factors that could have influenced outcomes.
Given the retrospective design and limited sample size, formal statistical adjustment for all these variables was not feasible. However, we have now expanded the Discussion (pages 11-12, lines 305-309) to address this. In our institution, occluder device selection for LAAC procedures is performed randomly rather than according to patient or anatomical characteristics. This approach helps to minimize selection bias and potential confounding arising from device type.
Comments 4:
Post-procedural antithrombotic therapy differed significantly between groups. Given the centrality of bleeding and thromboembolism outcomes, this heterogeneity should be addressed with stratified or sensitivity analyses.
Response 4
We thank the reviewer for this important observation. We acknowledge that post-procedural antithrombotic therapy differed among groups and may influence both bleeding and thromboembolic outcomes. However, due to the limited number of events and small subgroup sizes, further stratified or sensitivity analyses would lack statistical robustness. This limitation has been explicitly acknowledged in the revised Discussion (page 12, lines 323-333), where we note that variability in antithrombotic management represents a potential confounder.
Comments 5:
The composite endpoint includes all-cause mortality, which is highly influenced by comorbidities unrelated to thromboembolic protection. The authors should provide cause-specific mortality breakdowns and consider presenting stroke/systemic embolism separately.
Response 5
We appreciate the reviewer’s insightful comment. Unfortunately, the data on mortality is retrieved from the Ministry of the Interior. In most cases, the specific cause of mortality was not consistently available for all patients, preventing a reliable cause-specific mortality analysis. However, we have described stroke and systemic embolism outcomes separately to better reflect thromboembolic events independent of overall mortality (page 9, lines 245-250).
Reviewer 2 Report
Comments and Suggestions for Authors
The LAA oclussion is a fast-growing procedure that fills the gap for those patients with atrial fibrillation that need embolic events prevention and cannot take anticoagulation therapies for many clinical reasons. There's a reduced number of this patient that already have a thrombus in their atrial appendage in the moment of the deployment. As it's explained in the discussion the problem is there are very few patients with atrial thrombus in their sample and the expected incidence of the primary events is quite low so its very difficult to demonstrate differences between the two groups and you cannot make a multivariate statistical analysis to exclude possible confounding factors. Another striking data is the only use cerebral protection system (CPS) in 24% of this patients. Although there is not a standarized protocol for this procedure and the use of CPS in other settings such as transaortic valve implantation had suffered serious drawbacks maybe it's the most sensible approach. To sum up maybe the would need more patients to test their hypothesis.
Author Response
Comments 1:
The LAA oclussion is a fast-growing procedure that fills the gap for those patients with atrial fibrillation that need embolic events prevention and cannot take anticoagulation therapies for many clinical reasons. There's a reduced number of this patient that already have a thrombus in their atrial appendage in the moment of the deployment. As it's explained in the discussion the problem is there are very few patients with atrial thrombus in their sample and the expected incidence of the primary events is quite low so its very difficult to demonstrate differences between the two groups and you cannot make a multivariate statistical analysis to exclude possible confounding factors. Another striking data is the only use cerebral protection system (CPS) in 24% of this patients. Although there is not a standarized protocol for this procedure and the use of CPS in other settings such as transaortic valve implantation had suffered serious drawbacks maybe it's the most sensible approach. To sum up maybe the would need more patients to test their hypothesis.
Response 1:
We thank the reviewer for their thoughtful and comprehensive comments. We fully agree that left atrial appendage closure (LAAC) represents an important therapeutic option for patients with atrial fibrillation who are not suitable candidates for long-term anticoagulation. As the reviewer correctly notes, patients presenting with a pre-existing left atrial appendage thrombus constitute a small and highly selected subgroup. Consequently, the limited number of such patients and the overall low incidence of primary events restrict the statistical power to detect significant differences between groups and preclude a meaningful multivariate analysis. This limitation has been further emphasized in the revised limitations (page 14, lines 365-368).
Regarding the use of cerebral protection systems (CPS), we acknowledge that their use was limited in our cohort. However, the rate of CPS utilization in our study is comparable to that reported in other studies in the field (e.g., Tarantini et al.). As noted, there are currently no standardized recommendations for CPS use during LAAC, and the experience from other structural interventions such as transcatheter aortic valve replacement (TAVR) remains mixed. In the PROTECTED TAVR trial (NEJM 2022), among patients with aortic stenosis undergoing transfemoral TAVR, the use of cerebral embolic protection did not significantly reduce the incidence of periprocedural stroke. Similarly, data from the STS/ACC TVT Registry demonstrated only a modest, borderline significant reduction in the composite endpoint of stroke associated with death or discharge to a non-home location, with no significant differences in the adjusted rates of in-hospital stroke or non-disabling stroke.
In addition, routine use of CPS may increase procedural risk. The requirement for an additional arterial access, catheter manipulation within the aortic arch, and prolonged procedural time may contribute to higher rates of periprocedural complications. Nevertheless, we agree that CPS may represent a reasonable and potentially beneficial strategy in this high-risk patient population. In our study, the decision to use CPS was left to the discretion of the operator.
Finally, we concur that larger studies are warranted to validate our findings and to further assess the safety and efficacy of LAAC in patients with left atrial appendage thrombus.
Round 2
Reviewer 1 Report
Comments and Suggestions for Authors
The authors have meticulously revised their manuscript. I have no further suggestions.
Author Response
Comment 1: The authors have meticulously revised their manuscript. I have no further suggestions.
Response 1:
We thank the reviewer for their positive feedback and for confirming that no further revisions are required.
Reviewer 2 Report
Comments and Suggestions for Authors
I agree with the changes. However in flow chart where you explain patient selection it's reported that the patients whose LAA had been close using LAMBERT had been excluded.I think you meant Lambre.
So far I have no more comments
Author Response
Comment 1:
I agree with the changes. However in flow chart where you explain patient selection it's reported that the patients whose LAA had been close using LAMBERT had been excluded. I think you meant Lambre. So far I have no more comments
Response 1:
We thank the reviewer for noticing this typo. We have corrected it, and the term Lambre now appears correctly in the flow chart (page 22, line 413).